# Surface Morphology and Mechanical Properties of Polyether Ether Ketone (PEEK) Nanocomposites Reinforced by Nano-Sized Silica (SiO_2_) for Prosthodontics and Restorative Dentistry

**DOI:** 10.3390/polym13173006

**Published:** 2021-09-05

**Authors:** Ahmed Abd El-Fattah, Heba Youssef, Mohamed Abdel Hady Gepreel, Rafik Abbas, Sherif Kandil

**Affiliations:** 1Department of Materials Science, Institute of Graduate Studies and Research, Alexandria University, El-Shatby, Alexandria 21526, Egypt; rafik.abbas@yahoo.com (R.A.); s.kandil@usa.net (S.K.); 2Department of Chemistry, College of Science, University of Bahrain, Sakhir P.O. Box 32038, Bahrain; 3College of Dentistry, Arab Academy for Science, Technology and Maritime Transport (AASTMT), El-Alamein 51718, Egypt; heba.yousse@aast.edu; 4Department of Materials Science and Engineering, Egypt-Japan University of Science and Technology (E-JUST), New Borg El-Arab City 21934, Egypt; geprell@yahoo.com

**Keywords:** PEEK, nano-silica, nanocomposites, prosthodontics, surface properties, elastic modulus, microhardness, flexural strength, compression molding

## Abstract

In the field of orthopedics and traumatology, polyether ether ketone (PEEK) serves a significant role as a suitable alternative to traditional metal-based implants like titanium. PEEK is being used more commonly to replace traditional dental products. For bonding with various adhesive agents and preserved teeth, the surface alteration of PEEK was investigated. The aim of this research was to understand how different types and contents of nano-sized silica (SiO_2_) fillers influenced the surface and mechanical properties of PEEK nanocomposites used in prosthodontics. In this work, PEEK based nanocomposites containing hydrophilic or hydrophobic nano-silica were prepared by a compression molding technique. The influence of nano-SiO_2_ type and content (10, 20 and 30% wt) on surface properties of the resultant nanocomposites was investigated by the use of scanning electron microscopy (SEM), energy-dispersive X-ray spectroscopy (EDX), surface roughness analysis, and contact angle measurement. The crystalline structures of PEEK/SiO_2_ nanocomposites were examined by X-ray diffraction (XRD) spectroscopy. Mechanical properties were measured by microhardness, elastic compression modulus, and flexural strength. All nanocomposites showed increased surface roughness compared to pure PEEK. SEM images revealed that nanocomposites filled with low content hydrophobic nano-SiO_2_ showed uniform dispersion within the PEEK matrix. The introduction of 10 wt% of hydrophobic nano-SiO_2_ to the PEEK matrix improved elastic modulus, flexural strength, and microhardness, according to the findings. The addition of nano-SiO_2_ fillers in a higher weight percentage, over 10%, significantly damages the mechanical characteristics of the resultant nanocomposite. On the basis of the obtained results, PEEK/SiO_2_ nanocomposites loaded with low content hydrophobic nano-SiO_2_ are recommended as promising candidates for orthopedic and prosthodontics materials.

## 1. Introduction

Prosthodontics is a major branch of dental medicine. It has become the main driver of dental treatment and the centerpiece of dental science with the rise of human welfare, standard of living, and the increased level of oral health awareness. Prosthodontics mostly manages dental defects and rehabilitation after tooth loss (such as crowns, dentures and lays), as well as extending the uses of artificial prostheses for periodontal disease, temporomandibular joint disease, and maxillofacial tissue defects [1,2]. Prosthodontic dental materials can be classified into three categories: metals, ceramics and resins. The qualities of these materials are important in the construction of dental prostheses, which are in direct contact with the oral mucosa and under long-term use in the oral environment. Therefore, the dental materials must have comprehensive properties and good biological activity to function properly [3,4,5].

Dental materials should have high mechanical strength, toughness, higher fatigue strength, high elastic modulus, low thermal and electrical conductivity, good castability, and low shrinkage deformation. Chemical stability is also needed, such as resistance to corrosion, breakage, and the effects of aging. The colors of dental materials can be formulated and should keep long-term stability. A good oral material should have adequate biocompatibility and safety, and be biofunctional [6,7,8]. However, due to the nature of the material itself, its continued use for long period in moist environment, a variety of problems, such as pigment coalescence, color change, and aging fractures, do occur [9,10].

In recent years, advances in nanomaterials and technology have captured increased attention because of their unique structures and properties. Development of nanomaterials has strengthened many applications in medicine and dentistry [11,12,13]. Polymeric nanocomposites are a class of nanomaterials in which nanoscale particulates such as spherical inorganic minerals are dispersed within polymeric matrices [14,15,16,17]. Compared to pure polymers, polymeric nanocomposites are claimed to show markedly improved properties, such as modulus, strength, dimensional stability, electrical conductivity, barrier performance, solvent resistance, biocompatibility, low plaque affinity, good aesthetics, and characteristics close to dental structure depending on type and content of the nanofiller particles used [18,19].

Polyether ether ketone (PEEK) is an emerging kind of thermoplastic engineering plastic [20]. Modifications and alterations of PEEK have been extensively studied thanks to its wide range of applications in the areas of selective laser sintering, dehumidification, nanofiltration membranes, fuel cells, and biomedical devices [21,22,23,24]. PEEK is a semi-crystalline colorless polymer with outstanding mechanical and thermal properties. Excellent thermal stability PEEK materials have a melting point (*T*_m_) of 343 °C and a glass transition temperature (*T*_g_) of 143 °C. PEEK can be processed using a variety of commercial techniques, including injection molding, extrusion molding, compression molding, and additive manufacturing at temperatures between 350 and 420 °C. Moreover, PEEK is nontoxic, and its sterilization efficiency is excellent. It can be repeatedly sterilized using conventional methods such as those employing steam, gamma radiation, and ethylene oxide, without evident degradation of the mechanical properties [25].

PEEK possesses particularly high durability and firmness, mainly in terms of fatigue and its strength, which is equivalent to alloy constituents [26]. One of the important characteristics of PEEK is its reduced elastic modulus, which ranges between 2 and 6 GPa and efficiently prevents the pressure sheltering influence [27]. Since its elastic modulus is much more related to that of compact skeletal mass, PEEK has emerged as an operational and suitable substitute to traditional implantation such as titanium in the fields of orthopedics [28] and traumatology [29]. PEEK has also been intended for prosthodontic use to produce prosthetic infrastructures and abutments for titanium-based implant systems. However, in those cases, it is essential that the material exhibits high mechanical strength, wear resistance and aesthetic features compatible with those of tooth tissues.

Considerable research interest has recently been directed toward developing PEEK nanocomposite biomaterials to improve various physical, mechanical and barrier properties to allow their application to restorative dentistry [30]. This could be achieved by the incorporation of spherical inorganic nanoparticles, such as nano silica (SiO_2_), into the PEEK matrix [31].

Bare SiO_2_ is characterized by its small particle size and large surface area. The surface of silica has three chemical groups of isolated hydroxy, hydrogen-bonded hydroxy, and siloxane groups. Thus, the surface is usually hydrophilic, even though the siloxane groups are hydrophobic. The hydrophilic surface of bare SiO_2_, however, can be rendered hydrophobic by reacting its surface hydroxyl groups with organofunctional groups, such as polydimethylsiloxane, dimethyldichlorosilane, and hexamethyldisilane [20].

Compression molding has been used to successfully fabricate PEEK composites reinforced with SiO_2_ nanofillers for industrial processes [32]. However, research into SiO_2_ nanofillers as PEEK reinforcement in the dental field is minimal.

Several researchers have examined whether surface pre-treatments would enhance the bond strength between PEEK and dental materials [33,34,35]. In fact, in dental therapy, pre-treatment with a silane binding agent significantly helps in achieving a reliable bond between an inorganic filler-filled dental prosthesis and resin cement [36]. The effect of SiO_2_ in PEEK on its bearing capacity to resin cement was investigated in an analysis. The results of that research analysis discovered that increasing the SiO_2_ content in PEEK improved the tensile bond strength [37]. As a result, further research is required to fully understand the influence of nanofiller particle form, structure, particle size, total volume, and coating in PEEK for dental applications.

The goal of the present research is to assess the influence of hydrophilic and hydrophobic nano-SiO_2_ fillers on the surface and mechanical properties of PEEK nanocomposites for prosthodontics and restorative dentistry, due to the lack of data on PEEK/SiO_2_ nanocomposite performance for dental applications.

## 2. Materials and Methods or Experimental

### 2.1. Materials

Semi-crystalline PEEK fine powder (VICTREX^®^ PEEK polymers, Victrex Technology Centre, Lancashire FY5 4QD, UK.) with average particle size 50 μm for compression molding was used as the polymer matrix. The density of PEEK is 1.3 g/cm^3^, and its melt viscosity is 350 Pa.s. Two types of amorphous nano-SiO_2_ particles (NanoTech Egypt Co., Giza, Egypt) were used as filler materials. The bare hydrophilic nano-SiO_2_ had an average particle size of 29 ± 4 nm, while the chemically modified hydrophobic nano-SiO_2_ (treated with trimethyl chlorosilane) had an average particle size of 14.5 ± 5 nm.

### 2.2. Preparation of PEEK/SiO_2_ Nanocomposites

Prior to mixing, PEEK powder and nano-SiO_2_ particles were dried overnight in a vacuum oven at 120 °C to guarantee sufficient elimination of moisture. Then they were ball-milled mixed in a planetary ball mill (Emax, Retsch GmbH, Haan, Germany) at 25 °C and 400 rpm for 2 h. The PEEK/SiO_2_ nanocomposites were fabricated using a compression molding process. The as-milled dried powder was filled in tool steel die with 10 mm diameter. The powder was compressed at room temperature under a pressure of 35 MPa for 2 min. After cold compaction, the powder was heated to 410 °C, while applying a low cavity pressure of about 2 MPa. Once the system reached the set temperature, it was held at constant temperature and pressure for 10 min to establish homogeneity within the melt. Following this, the system was permitted to cool down to room temperature under a pressure of 20 MPa. Finally, the mold was opened and the samples were taken out. Custom-designed mold tooling was used to produce the samples for mechanical testing. Table 1 illustrates the precise composition of the prepared PEEK/SiO_2_ nanocomposites used in this analysis. PK was used to identify PEEK polymers in this study. Hydrophobic and hydrophilic nano-SiO_2_ particles were assigned the codes BS and LS, respectively, followed by a number indicating the weight percentage of nano-SiO_2_ particles. For example, the PEEK/SiO_2_ nanocomposite with 10 wt% hydrophobic nano-SiO_2_ particles, was coded as PKBS-10.

### 2.3. Characterization Methods

#### 2.3.1. X-ray Diffraction (XRD)

Crystal structural analyses of the pure PEEK and PEEK/SiO_2_ nanocomposites were performed by powder XRD measurements using a diffractometer (XRD-7000, Shimadzu, Japan). The X-ray beam was Cu-Kα radiation (λ = 0.1542 nm), operated at 40 kV and 30 mA. The XRD pattern was recorded in the 2θ range from 10° to 50° with a scanning rate of 5° per min. The crystalline phase was identified and compared to the literature as well as the International Center for Diffraction Data (ICDD) for PEEK.

#### 2.3.2. Scanning Electron Microscopy (SEM)

A scanning electron microscope (SEM) was used to assess the surface morphology of the desired samples. For this purpose, SEM uses a JEOL instrument (JSM-5300, Tokyo, Japan) which was operated at 25 keV. Prior to SEM imaging, the samples were ultrasonically washed for 30 s and sputter-coated with gold to a thickness of 400 Å in a sputter-coating device (JFC 1100 E).

#### 2.3.3. Energy-Dispersive X-ray Spectroscopy (EDX)

The presence of silica on each nanocomposite surface was determined by energy dispersive X-ray (EDX) microanalysis attached to the SEM. Analysis was performed on uncoated samples at 15 kV for 60 s.

#### 2.3.4. Surface Roughness Analyses

The surface roughness (Ra) of the desired samples was examined by an optical profilometer (MarSurf PS1, Mahr GmbH Göttingen, Germany). Four different locations perpendicular to the surface on each sample were recorded. Mean Ra values were statistically analyzed and used as the final Ra score for each sample.

#### 2.3.5. Contact Angle Measurement

Water contact angle experiments in a goniometer digital (RAMÉ-hart Model 190-F2, Succasunna, NJ 07876, USA) had been used to determine the surface hydrophilicity of the samples. The static sessile drop technique was performed using the video contact angle method. To measure the average contact angle and standard deviation, at least seven stabilized contact angles from different sites in each sample were obtained.

#### 2.3.6. Microhardness Measurement 

Microhardness of the samples was measured as Vickers hardness number (VHN) (Wolpert micro-Vickers tester, Wolpert Wilson Instruments, division of Instron Deutschland GmbH, Aachen, Germany). The indentations were made using a diamond pyramid micro-indenter with a 136° angle between the opposing faces under a load of 200 N applied for a dwell time of 10 s.

#### 2.3.7. Mechanical Tests

Compression analysis was carried out by using a universal testing machine (AG-IS 100KN, Shimadzu Corporation, Kyoto, Kyoto Prefecture, Japan). The test was performed with a cylindrical sample of dimensions (10 mm diameter × 6 mm height) at a crosshead speed of 1.0 mm/min until the specimen failed. Flexural properties were measured using a three-point bending test method in the same universal testing machine. The test was carried out with a rectangular bar sample of dimension (80 mm × 6 mm × 6 mm) at a crosshead speed of 1.0 mm/min at room temperature. A total of seven cases were analyzed for each sample at room temperature. Seven samples were tested for each material group (PEEK, PKBS or PKLS) to obtain the average value.

### 2.4. Statistical Analysis

Statistical Package for the Social Sciences (SPSS) software was used to analyze the results (IBM Corp.: Armonk, NY, US). The Kolmogorov–Smirnov (K–S) test was used to conduct parametric tests, and the results showed that the data were normally distributed. Multiple comparison analyses were performed using variance (ANOVA) analysis with *p* = 0.05, Tukey post hoc test, and Student’s *t*-test.

## 3. Results and Discussion

### 3.1. Fabrication of Nanocomposites

PEEK/SiO_2_ nanocomposites were successfully fabricated by dry mixing of PEEK (polymer matrix) and diverse nano-SiO_2_ particles (inorganic nanofiller) using high energy ball milling followed by a compression molding process. Ball milling technique was used to disperse nano-SiO_2_ particles into the PEEK matrix due to the excellent deformability of thermoplastic polymers. In addition, ball milling was shown to improve the mechanical properties of the polymer by reducing the particle size of pure PEEK from a millimeter to a micrometer scale (~5 μm). It is well known that PEEK has a good resistance to most organic solvents except concentrated sulfuric acid (95–98%) and methyl sulfonic acid [37]. As a result of the poor solubility of the PEEK in organic solvents, it is more feasible to fabricate its nanocomposites through the compression molding technique.

In this work, the number of nano-SiO_2_ particles was varied at 10%, 20% and 30%. These values were selected because they represent an optimum performance of mechanical properties and the quality of interface between the nanoparticles and polymer matrix [28]. Nevertheless, concentrations exceeding the 30% threshold were discounted to avoid the strong repulsion and attraction forces of nano-SiO_2_ particles, which in turn may deteriorate the overall mechanical properties of the composite and hinder the adhesion at the interface between the matrix and the nanofillers [37,38,39,40].

### 3.2. Structural Analysis

Figure 1 shows the XRD diffraction patterns of the pure PEEK and PEEK/SiO_2_ nanocomposites (PKBS and PKLS groups) loaded with diverse nano-SiO_2_ contents (10, 20, and 30 wt%). XRD of both hydrophobic and hydrophilic nano-SiO_2_ particles did not show any sharp Bragg peaks, except a broad peak between 15° and 25°, signifying that they have an amorphous structure. The pure PEEK and its nanocomposites crystallize mostly in the form of an orthorhombic crystal structure. The pure PEEK exhibits diffraction peak positions (2θ) of about 18.61°, 20.53°, 22.45°, and 28.62°, corresponding to diffraction planes of (110), (111), (200), and (211).

Evidently, apart from those of pure components, no new diffracting peaks were detected in the diffraction pattern of PEEK/SiO_2_ nanocomposites. Moreover, all nanocomposite samples showed the same XRD patterns with decreasing peak intensities in proportion to the nano-SiO_2_ content. The results indicated that the increase in both hydrophobic and hydrophilic nano-SiO_2_ content decreases the crystallinity of the PEEK matrix. Similarly, in a previous study conducted on the crystallization behavior of the nano-SiO_2_ filled PEEK composites, it was concluded that the inclusion of the 15 nm SiO_2_ particles would significantly decrease the crystallinity of the PEEK matrix by about 15% under isothermal crystallization, due to the hindrance of mobility [39]. Experimental evidence indicated that nano-SiO_2_ has little effect on the degree of crystallinity and that it does not act as a nucleating agent [40]. It has been shown that, for the given number of nanoparticles, the polymer crystallization was reduced by impeding the arrangement of molecular chains for the formation of the lamellae [37].

### 3.3. Morphological Observation

SEM micrographs of the pure PEEK and PEEK/SiO_2_ nanocomposites loaded with different nano-SiO_2_ contents (10, 20, and 30 wt%) were captured to examine the exact microstructure as shown in Figure 2. As displayed in Figure 2a, the pure PEEK micrograph revealed a relatively homogenous, smooth, and uniform surface. The effect of incorporation of hydrophobic nano-SiO_2_ on the morphology of the nanocomposites is illustrated in Figure 2b–d. The addition of low hydrophobic nano-SiO_2_ content (10 wt%) in the nanocomposite showed uniform dispersion within the polymer matrix, leaving a relatively smooth surface (Figure 2b). On the contrary, a clear nano-SiO_2_ agglomeration and increased surfaces roughness were observed in the nanocomposites containing high hydrophobic nano-SiO_2_ content (30 wt%).

On the other hand, the SEM micrographs of the PEEK/SiO_2_ nanocomposites based on different hydrophilic nano-SiO_2_ contents (10, 20, and 30 wt%), as presented in Figure 2e,f, exhibited poor dispersibility and weak interfacial adhesion of hydrophilic nano-SiO_2_ with the polymer matrix. Moreover, the PEEK/SiO_2_ nanocomposites based on hydrophilic nano-SiO_2_ (PKLS group) exhibited a rougher surface than their corresponding PEEK/SiO_2_ nanocomposites based on hydrophobic nano-SiO_2_ (PKBS group) counterparts.

These results might be due to the additions of nano-SiO_2_ particles to the PEEK matrix, leading to various nano-SiO_2_ particle–particle and nano-SiO_2_ particle–PEEK chain interactions, allowing formations of aggregates and agglomerates on the surface of PEEK nanocomposites [20,37]. Such agglomerations are less evident for PEEK/SiO_2_ nanocomposites containing hydrophobic nano-SiO_2_ particles, causing smoother surfaces compared with PEEK/SiO_2_ nanocomposites embedded with hydrophilic nano-SiO_2_ particles.

### 3.4. Compatibilization of Hydrophobic Polymer and Nanofiller

The quality of the filler-matrix interface is significant for the application of inorganic filler particles as reinforcing materials in polymer matrices. The properties of nanocomposites depend on all of the individual components and on their compatibility. In general, homogeneous and uniform distribution of nanofiller particles in polymer matrices is extremely crucial for the improvement of physicochemical properties and mechanical characteristics of polymer matrix nanocomposites (PMNCs). Therefore, the poor distribution of nanoparticles in polymer matrices causes potential problems in the fabrication of PMNCs [37,38]. It is generally acknowledged that the aggregation of particles is highly dependent on their dispersion within the polymer matrix. The increase in the degree of particle dispersion results in decreasing particle aggregation.

EDX analysis was conducted to examine the dispersion of nano-SiO_2_ on the surface of the PEEK polymer matrix. Figure 3 shows the presence of chemical elements on the surface of pure PEEK as well as on the surface of various PEEK/SiO_2_ nanocomposites. In these patterns, there is no Si present on pure PEEK (Figure 3a). EDX analysis of PEEK/SiO_2_ nanocomposites containing 10 wt% hydrophobic nano-SiO_2_ as presented in Figure 3b illustrates more efficient interactions and higher compatibility between PEEK chains and hydrophobic nano-SiO_2_, leading to a lower presence of Si at these composites’ surfaces. However, hydrophilic nano-SiO_2_ particles (Figure 3e–g) containing hydroxyl groups have a strong tendency to agglomerate due to their incompatibility with the matrix, leading to migration to the surface of such composites. This finding is also further supported by previous reported data [41,42].

Based on the SEM observations and EDX analysis, a schematic presentation of possible interactions and dispersion of nano-SiO_2_ particles in the PEEK matrix is presented in Figure 4. The presence of hydroxyl groups on the surface of hydrophilic nano-SiO_2_ particles increases the particles’ interactions, through hydrogen bonding, which cause particle aggregation and network formation–potentially adversely influencing the nanoparticles’ distribution in the PEEK matrix [20]. In addition, the strength of interfacial interactions between the PEEK and nano-SiO_2_ particles is the most determining factor affecting the properties of the obtained nanocomposites. It is possible that hydrophilic nano-SiO_2_ particles make interfacial interactions and bond with hydroxyl end groups of PEEK chains. On the other hand, surface chemical modification is one of the various methods to improve the compatibility between the hydrophobic PEEK and hydrophilic nano-SiO_2_. Consequently, a good dispersion may be achieved by using hydrophobic nano-SiO_2_, due to its uniform incorporation within the hydrophobic PEEK matrix.

### 3.5. Surface Roughness (Ra)

Average values of surface roughness for the pure PEEK and PEEK/SiO_2_ nanocomposites loaded with diverse nano-SiO_2_ contents (10, 20, and 30 wt%) are summarized in Table 2. The results indicated that the surface roughness values increase with the increase of the nano-SiO_2_ contents in the nanocomposites. For example, the addition of 30 wt% hydrophilic nano-SiO_2_ markedly increased the surface roughness value of the pure PEEK from 1.43 to 3.32 μm (130%) in the nanocomposite. Furthermore, the roughness values are higher for the PEEK/SiO_2_ nanocomposites embedded with hydrophilic nano-SiO_2_ particles compared the hydrophobic ones. The approach of using additives to modify surface and bulk properties of performance polymers is well-known [41]. Generally, the ability of additives to migrate to the surface is defined by factors such as particle size, composition, end-group functionalities, molecular architecture, and concentration. Besides, the careful selection of additives with proper functionalities provides significant control over the hydrophilicity or hydrophobicity of modified surfaces while retaining bulk properties [42]. The PEEK and nano-silica particles have some interfacial interactions within the PEEK matrix. However, it appears that these interactions are not strong enough to enable a good dispersion of the hydrophilic nano-SiO_2_ particles within the PEEK matrix [43]. The formation of hydrogen bonds between the surface hydroxyl groups of the hydrophilic nano-SiO_2_ particles may be responsible for particle aggregation within the matrix [44].

PEEK chains are more compatible with hydrophobic nano-SiO_2_ particles compared to hydrophilic ones. This is a major problem in all nanocomposites, where nanoparticles, because of their high surface area, tend to create agglomerates [40]. This is more observable for PEEK/SiO_2_ nanocomposites containing hydrophilic nano-SiO_2_ particles in this study. Thus, the presence of these particles results in increased surface roughness, which is in good agreement with the SEM micrographs.

### 3.6. Contact Angle Measurement 

The variation of contact angle against different type and contents of nano-SiO_2_ in the nanocomposites is given in the Table 2. In case of PEEK/SiO_2_ nanocomposites loaded with hydrophobic nano-SiO_2_, the addition of 10 wt% of the hydrophobic nano-SiO_2_ increases the contact angle value by 30% compared to the pure PEEK. Nevertheless, higher contents (20 and 30 wt%) reveal a decline in the contact angle values. This may be due to the presence of an excess of hydrophobic nano-SiO_2_ on the surface of the nanocomposites [20,37]. Regarding the PEEK/SiO_2_ nanocomposites loaded with hydrophilic nano-SiO_2_, it can be observed that the contact angle increases with an increase in hydrophilic nano-SiO_2_ content in the nanocomposites. The increase in contact angle value can be compared with the increase in surface roughness of the PEEK surface. It has been shown that the contact angle increases with the increase in the roughness of the surface. These results show that the addition of the nano-SiO_2_ particles can alter the hydrophobic character of the pure PEEK matrix, and the contact angle can be changed markedly via changing the content of the additive.

### 3.7. Microhardness

The results of the microhardness test for the pure PEEK and PEEK/SiO_2_ nanocomposites are presented in Figure 5. The microhardness values increase significantly (*p* < 0.05) when low contents (10 and 20 wt%) of hydrophobic nano-SiO_2_ were incorporated in the nanocomposite formulation. For example, the addition of 10 wt% hydrophobic nano-SiO_2_ increased the microhardness value from 38 to 57 Kg/mm^2^, that is, 50% higher than pure PEEK. This result could be due to the homogeneity and uniformity of the particles’ distribution which gives rise to good adhesion between the particles and the PEEK matrix. Additionally, a decrease in interparticle distance, as particle loading in the matrix increased. led to an increase in PEEK matrix tolerance to indentation. Nanoparticles in the matrix are more alike than microparticles in the matrix for a given volume fraction, so nanoparticles can resist the indentation in the matrix more strongly [45]. On the other hand, a further increase in the hydrophobic nano-SiO_2_ content (30 wt%) showed a statistically insignificant decrease in the microhardness value (*p* > 0.05). The reason for this behavior may be attributed to the agglomeration of the nanoparticles and the lack of uniform dispersion.

The same trend of microhardness values was also observed in the PEEK/SiO_2_ nanocomposites filled with hydrophilic nano-SiO_2_ particles. It is worth noting that nanocomposites filled with hydrophilic nano-SiO_2_ particles had significantly lower microhardness values compared to those filled with hydrophobic nano-SiO_2_ particles.

These results may be endorsed by the alteration in particle size compatibility of the fillers used. The hydrophobic nano-SiO_2_ has a smaller particle size (14.5 ± 5 nm), better microhardness values compared to the hydrophilic nano-SiO_2_ (29 ± 4 nm), and more level dispersal in the PEEK matrix, therefore changing orientation of the polymer chains. Consequently, this leads to increased surface free energy of the resulting nanocomposite. Indeed, the surface properties of this nano-SiO_2_ make it more compatible with PEEK molecular chains, resulting in raised total and polar surface free energies as well as microhardness [46].

Likewise, in similar research using SiO_2_ nanoparticles with sizes ranging from 15 to 30 nm, the composites with finer nanoparticles witnessed a significant and linear increase in hardness, even at the maximum SiO_2_ content of 10% by weight. The finer 15 nm particles are more evenly distributed, resulting in a continuously increasing hardness [47].

### 3.8. Mechanical Properties

The compression elastic modulus and flexural strength of PEEK and its nanocomposites, fabricated by adding different contents of hydrophobic nano-SiO_2_ and hydrophilic nano-SiO_2_, respectively, are shown in Figure 6 and Figure 7. The addition of 10 wt% hydrophobic nano-SiO_2_ showed a statistically significant increase in the compression elastic modulus by 40% compared to pure PEEK. However, there was a significant decrease by 5% and 65% in compression elastic modulus upon addition of 20 and 30 wt% hydrophobic nano-SiO_2_, respectively, compared to pure PEEK.

As for the nanocomposites loaded with hydrophilic nano-SiO_2_, the same pattern was observed. The compression elastic modulus was significantly enhanced by 25.3% compared to pure PEEK in the 10 wt% nanocomposite. Nevertheless, there was a significant decrease by 34.1% and 124.1% in the compression elastic modulus in 20 and 30 wt% nanocomposite, respectively. The measured elastic modulus data of nanocomposites filled with hydrophobic nano-SiO_2_ appears significantly higher than the that of nanocomposites loaded with hydrophilic nano-SiO_2_, suggesting the effective enhancement of the filled hydrophobic nano-SiO_2_ particles.

Regarding the flexural strength, 10 wt% hydrophobic nano-SiO_2_ filled PEEK nanocomposite showed a significant increase of 33.9%, compared to pure PEEK, the trend of the increasing hydrophobic nano-SiO_2_ content until 20 wt%. However, there was a significant decrease by 37.5% and 116.5% in flexural strength for nanocomposites loaded with 20 and 30 wt% hydrophobic nano-SiO_2_, respectively.

As for the nanocomposites loaded with hydrophilic nano-SiO_2_, the flexural strength value decreased by 3.3% for 10 wt% nanocomposite compared to PEEK, but this decrease was found to be insignificant. However, there was a significant decrease of 12.7% and 191.79% in flexural strength nanocomposites loaded with 20 and 30 wt% hydrophilic nano-SiO_2_, respectively, compared to the pure PEEK. Comparing the PKBS group to their corresponding PKLS group, there was a significant increase in flexural strength between PKBS-10 and PKLS-10. Moreover, the flexural strength values of the hydrophobic nano-SiO_2_-filled PEEK nanocomposites are statistically significantly higher than those of the hydrophilic nano-SiO_2_.

In general, it has already been documented that adding fillers to PEEK will improve its mechanical properties [48,49], while the addition of 30 wt% of nano-SiO_2_ particles significantly decreased the mechanical properties. The incorporation of 30% vol of calcium silicate into PEEK resulted in a decrease of 20.84% in bending strength, according to a previously reported data [50].

The polarity imbalance between some of the hydrophilic nano-particle surfaces and the PEEK matrices, which resulted in poor dispersion of the nano-particles, could explain the higher elastic modulus of the hydrophobic nano-SiO_2_ filled groups over their respective hydrophilic nano-SiO_2_ filled groups [51]. Accordingly, the hydrophobic nano-SiO_2_ particles disperse better in the PEEK matrix because of the matching polarity.

## 4. Conclusions

The outcomes of the current research suggested that the addition of 10% hydrophobic nano-SiO_2_ to the PEEK polymer matrix resulted in an improvement of the elastic modulus, flexural strength, and microhardness. Despite the high mechanical properties of the 10% hydrophilic nano-SiO_2_ filled PEEK nanocomposite, compared to the pure PEEK, it is still significantly lower than the same weight percentage of hydrophobic nano-SiO_2_-filled PEEK nanocomposite. The incorporation of nano-SiO_2_ fillers in a higher weight percentage (20% and 30%) significantly damages the mechanical characteristics of the resultant nanocomposites. As a result of the findings, treated PEEK/SiO_2_ nanocomposites based on 10% hydrophobic nano-SiO_2_ might be ideal for prosthodontics and restorative dentistry. Hence, it can be considered a promising potential alternative to metals such as titanium and zirconium due to its biochemical composition and high-quality mechanical properties. Despite being widely used as a progenitor material in the spinal column, orthopedics, and sports medicine, it has yet to reach critical mass in dental practice. However, further long-term clinical research into PEEK polymer as a substitute material for conventional metals is necessary.

## Figures and Tables

**Figure 1 polymers-13-03006-f001:**
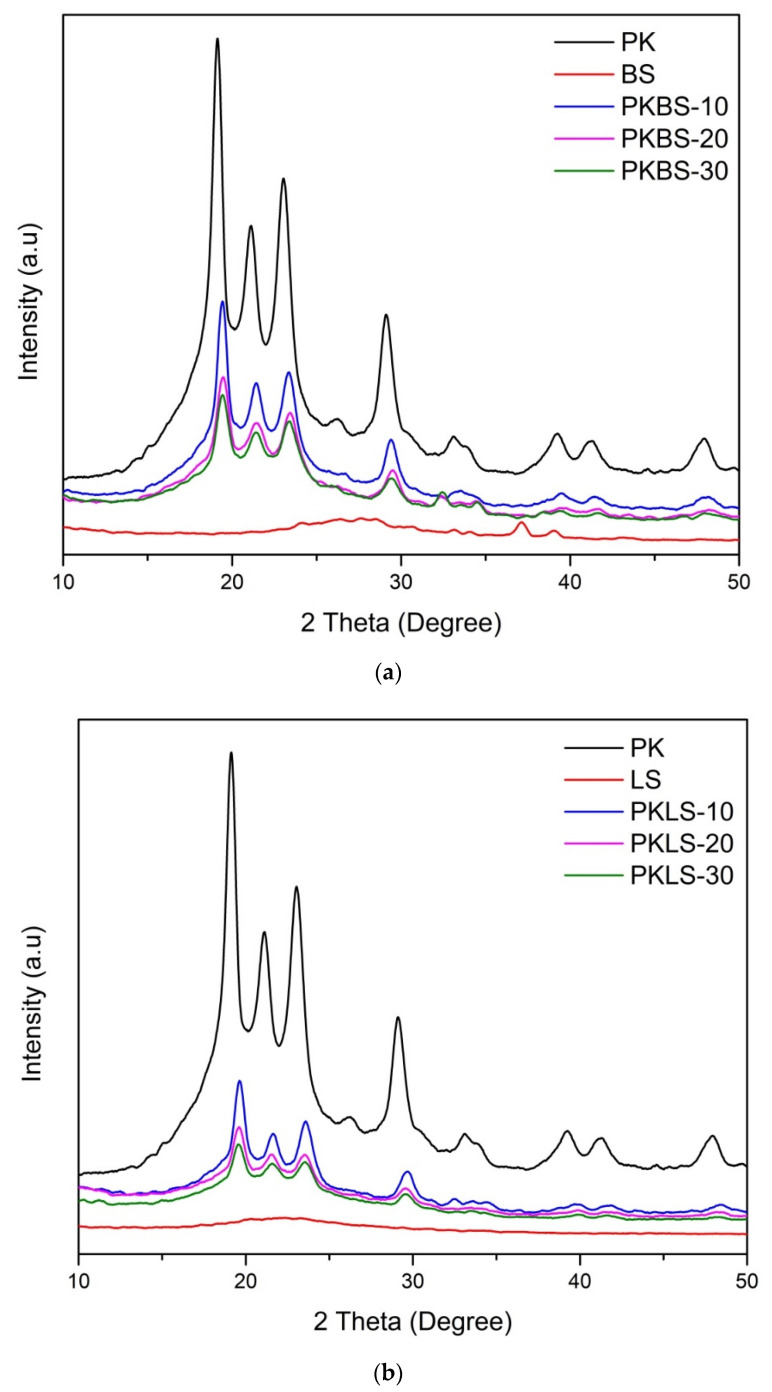
XRD patterns showing the effect of the nano-SiO_2_ fillers on the crystallinity of PEEK: (**a**) nanocomposites filled with hydrophobic nano-SiO_2_ and (**b**) nanocomposites filled with hydrophilic nano-SiO_2_.

**Figure 2 polymers-13-03006-f002:**
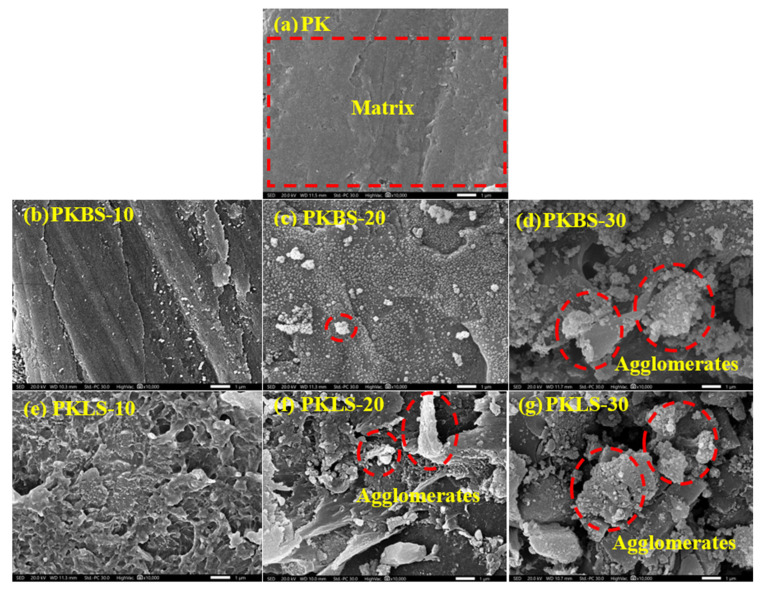
SEM micrographs showing the dispersion of the nano-SiO_2_ fillers in the PEEK matrix.

**Figure 3 polymers-13-03006-f003:**
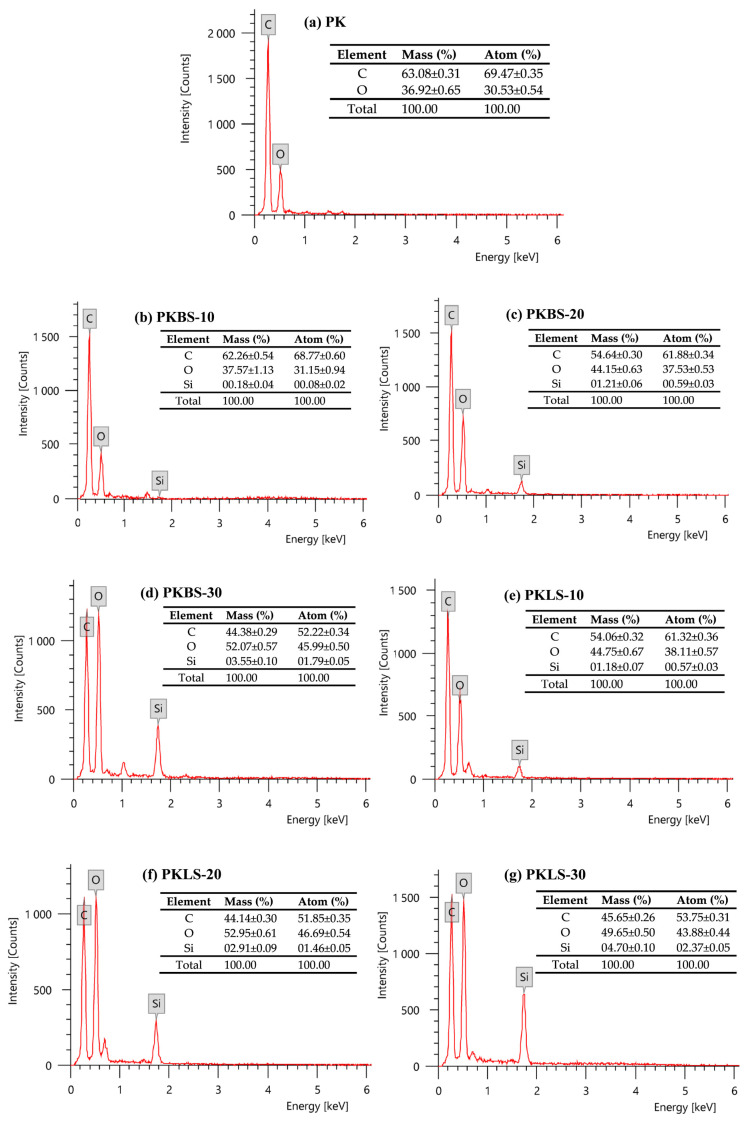
EDX analysis of pure PEEK as well as various PEEK/SiO_2_ nanocomposites.

**Figure 4 polymers-13-03006-f004:**
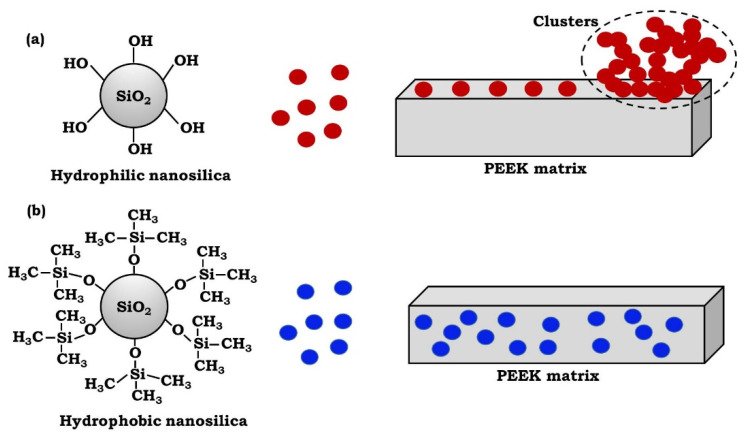
Schematic representation of the degree of dispersion and distribution of nano-SiO_2_ particles in the PEEK polymer matrix. (**a**) The hydrophilic nano-SiO_2_ particles are fused together into clusters with an irregular, chain-like geometry. (**b**) The hydrophobic nano-SiO_2_ particles exhibit good dispersion and distribution within the hydrophobic PEEK matrix.

**Figure 5 polymers-13-03006-f005:**
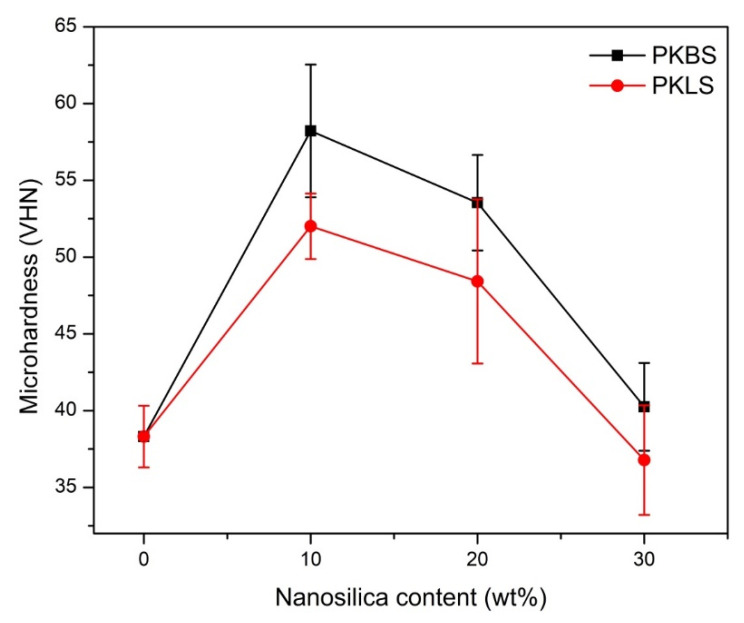
Mean value and standard deviations (mean ± SD) of microhardness for the PEEK/SiO_2_ nanocomposites as a function of the nano-SiO_2_ content. (*n* = 7).

**Figure 6 polymers-13-03006-f006:**
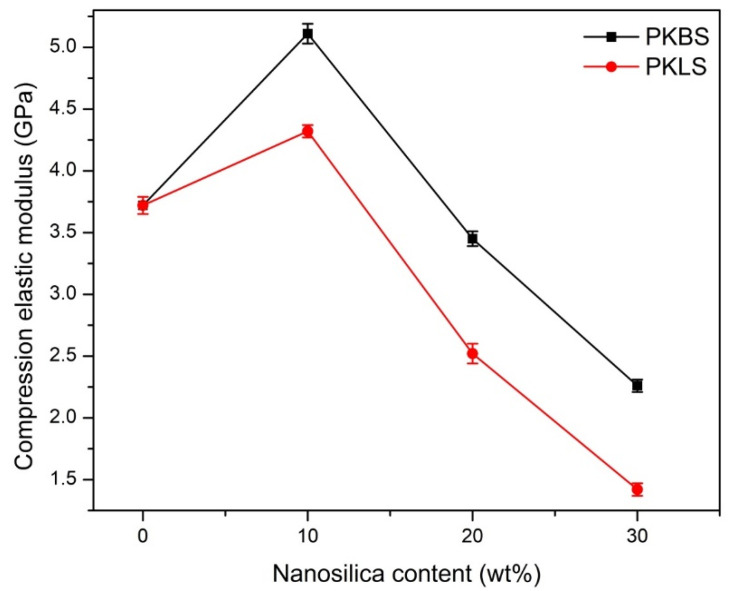
Mean value and standard deviations (mean ± SD) of elastic compression modulus for the PEEK/SiO_2_ nanocomposites as a function of the nano-SiO_2_ content. (*n* = 7).

**Figure 7 polymers-13-03006-f007:**
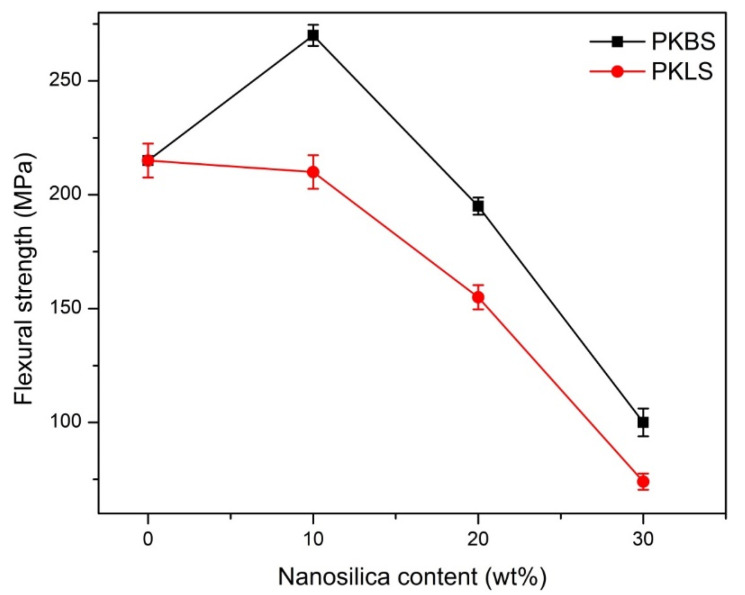
Mean value and standard deviations (mean ± SD) of flexural strength for the PEEK/SiO_2_ nanocomposites as a function of the nano-SiO_2_ content. (*n* = 7).

**Table 1 polymers-13-03006-t001:** PEEK/SiO_2_ nanocomposites formulations (percentage by weight).

Silica Content				
Code	Sample	PEEK (wt%)	BS (wt%)	LS (wt%)
PK	Unfilled PEEK	100	0	0
PKBS-10	PEEK/BS 10 wt%	90	10	0
PKBS-20	PEEK/BS 20 wt%	80	20	0
PKBS-30	PEEK/BS 30 wt%	70	30	0
PKLS-10	PEEK/LS 10 wt%	90	0	10
PKLS-20	PEEK/LS 20 wt%	80	0	20
PKLS-30	PEEK/LS 30 wt%	70	0	30

**Table 2 polymers-13-03006-t002:** Mean values and standard deviation (mean ± SD) for surface roughness and contact angle data of pure PEEK together with various PEEK/SiO_2_ nanocomposites.

Code	Sample (*n* = 7)	Surface Roughness (Ra) (μm)	Contact Angle (◦)
PK	Unfilled PEEK	1.45 ± 0.35	93.71 ± 1.52
PKBS-10	PEEK/BS 10 wt%	1.47 ± 0.23	122.40 ± 2.16
PKBS-20	PEEK/BS 20 wt%	2.03 ± 0.35	94.90 ± 1.10
PKBS-30	PEEK/BS 30 wt%	2.36 ± 0.32	60.59 ± 0.52
PKLS-10	PEEK/LS 10 wt%	1.52 ± 0.24	98.52 ± 1.75
PKLS-20	PEEK/LS 20 wt%	2.13 ± 0.16	113.10 ± 1.33
PKLS-30	PEEK/LS 30 wt%	3.32 ± 0.22	117.54 ± 1.07

## Data Availability

Not applicable.

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
