# Peer review of "Surface Morphology and Mechanical Properties of Polyether Ether Ketone (PEEK) Nanocomposites Reinforced by Nano-Sized Silica (SiO2) for Prosthodontics and Restorative Dentistry"

_polymers, 2021, doi:10.3390/polym13173006_

Round 1

Reviewer 1 Report

The scientific paper "Surface morphology and mechanical properties of polyether ether ketone (PEEK) nanocomposites reinforced by nano-sized silica (SiO2) for prosthodontics and restorative dentistry" aimed to understand how different types and contents of nano-sized silica (SiO2) fillers influenced the surface and mechanical properties of PEEK nanocomposites used in prosthodontics.

The manuscript is well written and presents a relevant subject for Dentistry. It can be considered that:

1) Correct the form of call in the text of the references according to the Polymers journal

2) The introduction is too long. Reduce by 25%.

3) The conclusion must be more concise and directly linked to the purpose of the study

4) Did the study have limitations? If so, insert at the end of the discussion.

Author Response

I appreciate the reviewer’s comments. Thank you for the reviewer for the useful and educative comments. The followings are my point-by-point responses:

1. Correct the form of call in the text of the references according to the Polymers journal

Response: Thank you to the reviewer for deferring our attention to this point. Therefore, the form of call in the text of the references has been corrected according to the Polymers journal.

2. The introduction is too long. Reduce by 25%

Response: I do agree with the reviewer’s comment, in turn, the introduction part has been reduced by 25%.

3. The conclusion must be more concise and directly linked to the purpose of the study

Response: I do agree with the reviewer’s comment, in turn, the conclusion has been modified accordingly.

4. Did the study have limitations? If so, insert at the end of the discussion

Response: No limitations were found in our study as all experiments were done in a controlled laboratory environment.

Reviewer 2 Report

  1. What was the rationale for the investigation of 10-30% of BS/LS? How was this range identified? What about lower and higher BS/LS? Justification and some discussion on this should be provided for the readers.
  2. The results section starts with literature analysis and not the description of the new results. Discussion on literature data should be minimized under the Results section.
  3. Section 3.1 under the Results have no new data but assumptions and a figure based on the literature. The authors should provide their own results and analysis here in order to support Figure 1.
  4. The provided SEM images are not informative. Different magnifications should be provided to complement the existing ones. It is unclear which part of the samples are shown. There is not much information that the readers can deduce from the images.
  5. Provide EDX elemental mapping for the nanocomposite material and the benchmark PEEK-only material for comparison purposes.
  6. PEEK is emerging, and its modifications and alterations are becoming a focal point in various fields. This should be briefly highlighted by mentioning the penetration of PEEK polymers in various fields including SLS (10.1016/j.matdes.2021.109510), dehumidification (10.1039/D1TA03690D), nanofiltration (10.1039/D0TA08194A), fuel cells (10.1016/j.jece.2021.105876).
  7. Some errors are given, e.g. Table 2 and Figures 4-6, but the error derivation is unclear. The captions should describe how many independently prepared samples were analyzed to get the results and the errors. Also, are these errors standard deviations? Were the samples independently prepared?
  8. The variations of microhardness of the PEEK/SiO2 nanocomposites as a function of the nano-377 SiO2 content is too large. Why are the error bars that large? Is it difficult to reproduce the materials with the same properties? Discussions on reproducibility should be added into the manuscript.
  9. The conclusion section should be more specific. Summarize the main quantitative results. What is the impact on the field? How the field has been advances? Besides the specific research outcome, the general implications also need to be mentioned.
  10. Some critical edge should be given. What are the limitations and drawbacks of the applied materials and methodologies?
